# H-Space Sparse Autoencoders

**Ayodeji Ijishakin**\*
University College London

**Ming Liang Ang**
University College London

**Levente Baljer**
King's College London

**Daniel Tan**
University College London

**Hugo Fry**
Independent

**Ahmed Abdulaal**
University College London

**Aengus Lynch**
University College London

**James Cole**
University College London

## Abstract

In this work, we introduce a computationally efficient method that allows Sparse Autoencoders (SAEs) to automatically detect interpretable directions within the latent space of diffusion models. We show that intervening on a single neuron in SAE representation space at a single diffusion time step leads to meaningful feature changes in model output. This marks a step toward applying techniques from mechanistic interpretability to controlling the outputs of diffusion models, further ensuring the safety of their generations. As such, we establish a connection between safety/interpretability methods from language modelling and image generative modelling.

## 1   Introduction

Diffusion models excel in myriad applications, including text-to-image synthesis, image-to-image translation, inverse problems, super-resolution, and image editing [1–5]. Despite significant advances in diffusion model capabilities, studies on model interpretability have yet to systematically understand how internal activations relate to specific outputs [6]. Such understanding is crucial for precise control over outputs, as it may mitigate safety risks like generating offensive or fraudulent images, imitating artists without consent, and perpetuating societal biases [7–10]. Conversely, progress has been made toward producing safer language models via a systematic understanding of their internal representations [11–13]. This has been achieved through a set of techniques collectively called mechanistic interpretability, which aim to reverse-engineer internal processes within neural networks [14–17].

By understanding the internals of neural networks through reverse engineering their internals, we can identify the neurons responsible for specific tasks, allowing for intervention and modification of activations when they might lead to unsafe outputs, thus improving safety. Researchers have identified sets of neurons within language models involved in model behaviour such as the induction task, indirect object identification, the greater than task, vision-language integration and grokking [12, 14, 15, 18, 19]. Inspired by these results, the present work seeks to repurpose these methods for diffusion models to provide a step toward steering their outputs toward safer generations.

Reverse engineering neural network internals requires decomposing models into discrete components representing distinct features. The most natural choice is a single neuron. However, neurons may activate in response to many disjoint features [13, 20, 21]. This property is called polysemanticity, and it has been posited by Elhage et al. [21] to occur because of *superposition*, where neural networks

---

\*Direct all correspondence to: ayodeji.ijishakin.21@ucl.ac.uk

1st Workshop on Safe Generative AI at NeurIPS 2024.

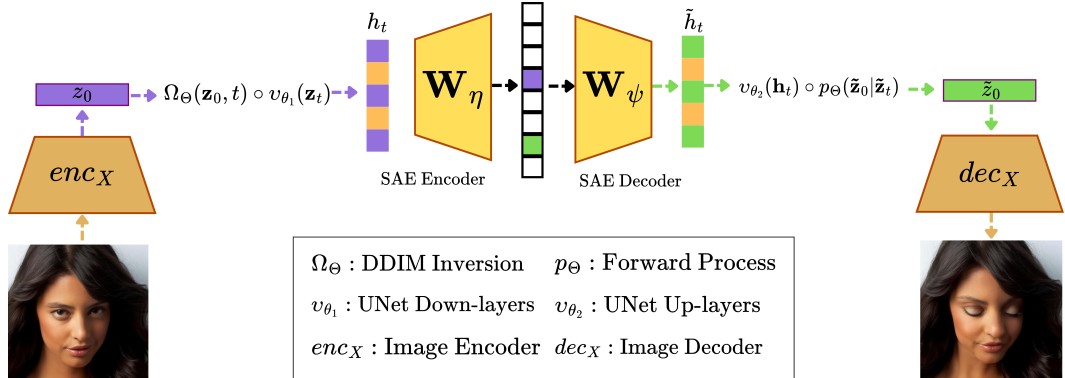

Figure 1: **The H-Space Autoencoder.** We take a pre-trained latent diffusion model, $\epsilon_\Theta^{(t)}(\mathbf{z}_t)$ and split it into its downward, $v_{\theta_1}(\mathbf{z}_t)$ and upward $v_{\theta_2}(\mathbf{z}_t)$ paths. We train our autoencoder ($\mathbf{W}_\eta$, $\mathbf{W}_\psi$), to reconstruct representations, $\mathbf{h}_t = v_{\theta_1}(\mathbf{z}_t)$ which are output by the downward path. Once trained, we can intervene on specific features by altering components of our Autoencoder representation to produce interpretable modifications to images.

learn more features than they have dimensions within a given layer. This implies that the learned basis, corresponding to distinct features, forms an overcomplete set of non-orthogonal vectors. The number of learned features would equal the layer's dimension if they were orthogonal.

Each neural network activation vector typically contains only a subset of features; for example, representations that correspond to an image of a dog will probably not contain features that relate to images of houses. To explain this phenomenon, the superposition hypothesis posits that activations are sparse linear combinations of distinct feature vectors. If so, it may be possible to reverse-engineer model features by finding vectors that reproduce neural network representations when sparsely combined. This is the motivation behind Sparse Autoencoders (SAEs), which build on the field of sparse dictionary learning by using neural networks to learn these feature vectors [13, 22, 23].

In this work, we extend the SAE framework to diffusion models. In particular, we examine *H-Space*, which is the intermediate latent space at the bottleneck of the U-Net architecture of diffusion models. Several works have highlighted its usefulness for representation learning in diffusion models [6, 24–30]. We show that we can find features that produce interpretable changes to model outputs when modified by learning vectors that, when sparsely combined, reproduce H-Space representations. We obtain these interpretable changes by modifying a single neuron in SAE representation space at a single timestep of diffusion model denoising. As such, our work is a step toward locating which neurons within diffusion models correspond to specific outputs, ensuring greater control and safety.

We have three main contributions. (1) We introduce the *Channel-Aware SAE*, which allows for SAEs to be applied to H-Space in a computationally efficient manner. (2) We introduce the first application of Sparse Autoencoders to the **internal** representations of a state-of-the-art Diffusion model. (3) We demonstrate that we learn image representations which are more robust and richer than existing methods.

## 2   Background

### 2.1   Diffusion Models

Denoising diffusion probabilistic models (DDPMs) are hierarchical latent variable models, where our latent variables, $\mathbf{x}_1, ..., \mathbf{x}_T$ are time-dependent and share the same dimension as our data $\mathbf{x}_0 \sim q(\mathbf{x}_0)$. To learn our data distribution, $q(\mathbf{x}_0)$, we express the joint distribution over all latents as a Markov chain, starting at a prior distribution $p(\mathbf{x}_T) = \mathcal{N}(\mathbf{x}_T; 0, \mathbf{I})$ and ending at our model distribution $p_\Theta(\mathbf{x}_0)$.

$$p_\Theta(\mathbf{x}_{0:T}) := p(\mathbf{x}_T) \prod_{t=1}^{T} p_\Theta^{(t)}(\mathbf{x}_{t-1}|\mathbf{x}_t) \tag{1}$$

$\Delta = 77$

Mouth

$\Delta = 84$

Weight

$\Delta = 77$

Female

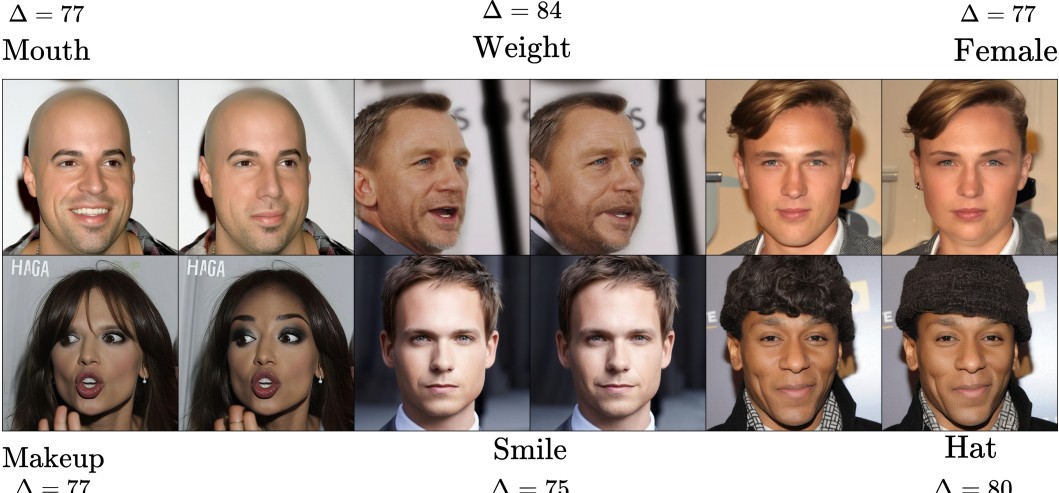

Makeup

$\Delta = 77$

Smile

$\Delta = 75$

Hat

$\Delta = 80$

Figure 2: **Results from interventining on HSpace Autoencoder representations.** Modifying a single neuron in HSoace-SAE space at a single timestep leads to localized feature changes. $\Delta$ denotes the number of neurons that change before and after our intervention. The total size of H-Space is 81920 dimensions.

This joint distribution, $p_\Theta(\mathbf{x}_{0:T})$, is known as the *reverse process*. It allows us to recover the probability of our data $p_\Theta(\mathbf{x}_0)$ by approximating $p_\Theta(\mathbf{x}_0) = \int p_\Theta(\mathbf{x}_{0:T})d\mathbf{x}_{1:T}$. Our approximate posterior, $q(\mathbf{x}_{1:T}|\mathbf{x}_0)$, is referred to as the *forward process*. The *forward process* iteratively corrupts our data through the addition of Gaussian noise, which simulates a diffusion process. It is pre-defined as another Markov chain of the form:

$$q(\mathbf{x}_{1:T}|\mathbf{x}_0) := \prod_{t=1}^{T} q(\mathbf{x}_t|\mathbf{x}_{t-1}), \quad q(\mathbf{x}_t|\mathbf{x}_{t-1}) := \mathcal{N}(\mathbf{x}_t; \sqrt{1-\beta_t}\mathbf{x}_{t-1}, \beta_t\mathbf{I}) \tag{2}$$

Where $\beta_1, ..., \beta_T$ are scalar values that control the variance at each time step, $t$. As such, DDPMs are a two-step procedure, our data $\mathbf{x}_0 \sim q(\mathbf{x}_0)$ is corrupted by the forward diffusion process, $q(\mathbf{x}_{1:T}|\mathbf{x}_0)$, and then denoised by the reverse process $p_\Theta(\mathbf{x}_{0:T})$ to obtain our original density. At time $t$, the approximate posterior is another Gaussian, $q(\mathbf{x}_t|\mathbf{x}_0) = \mathcal{N}(\sqrt{\alpha_t}\mathbf{x}_0, (1-\alpha_t)\mathbf{I})$, where $\alpha_t = \prod_{s=1}^{t}(1-\beta_s)$. Thus, via reparameterisation, the noised image at time $t$ can be expressed as $\mathbf{x}_t = \sqrt{\alpha_t}\mathbf{x}_0 + \sqrt{1-\alpha_t}\epsilon$ where $\epsilon \sim \mathcal{N}(\mathbf{0}, \mathbf{I})$. This allows us to simplify our objective by learning a model, $\epsilon_\Theta^{(t)}(\mathbf{x}_t)$ (typically a U-Net), which predicts the sampled noise, $\epsilon$, based on $\mathbf{x}_t$ and $t$. We may also condition this model, on auxiliary information, $\mathbf{c}$, such as a text prompt relating to our image $\mathbf{x}_0$. This admits the following objective:

$$L := \sum_{t=1}^{T} \mathbb{E}_{\mathbf{x}_0 \sim q(\mathbf{x}_0), \epsilon_t \sim \mathcal{N}(\mathbf{0}, \mathbf{I})} \left[ \|\epsilon_\Theta^{(t)}(\sqrt{\alpha_t}\mathbf{x}_0 + \sqrt{1-\alpha_t}\epsilon_t, \mathbf{c}) - \epsilon_t\|_2^2 \right] \tag{3}$$

To obtain samples from our model distribution, $p_\Theta(\mathbf{x}_0)$, we start at our final noise latent $p(\mathbf{x}_T)$ and iteratively obtain $p_\Theta(\mathbf{x}_{t-1}|\mathbf{x}_t)$, using the following generative process:

$$\mathbf{x}_{t-1} = \sqrt{\alpha_{t-1}} \underbrace{\left( \frac{\mathbf{x}_t - \sqrt{1-\alpha_t}\epsilon_\Theta^{(t)}(\mathbf{x}_t, \mathbf{c})}{\sqrt{\alpha_t}} \right)}_{\text{predicted } \mathbf{x}_0} + \underbrace{\sqrt{1-\alpha_{t-1}-\sigma_t^2}\epsilon_\Theta^{(t)}(\mathbf{x}_t, \mathbf{c})}_{\text{direction pointing to } \mathbf{x}_t} + \underbrace{\sigma_t\epsilon_t}_{\text{random noise}} \tag{4}$$

Where $\epsilon_t \sim \mathcal{N}(0, \mathbf{I})$. By setting $\sigma_t = \sqrt{\frac{1-\alpha_{t-1}}{1-\alpha_t}}\sqrt{\frac{1-\alpha_t}{\alpha_{t-1}}}$ for all $t$, we obtain the original stochastic DDPM generative process.

### 2.1.1 Denoising Diffusion Implicit Models

By setting $\sigma_t = 0$ for all $t$, the generative process becomes deterministic and transforms into a deep diffusion implicit model (DDIM). This is important, as it also allows for a deterministic mapping

| **Algorithm 1** Training | **Algorithm 2** Interventions |
|---|---|
| 1: **repeat** | 1: $\mathbf{x} \sim q(\mathcal{X})$ |
| 2:  $\mathbf{x} \sim q(\mathcal{X})$ | 2: $t = 0.7T$ |
| 3:  $t = 0.7T$ | 3: $\mathbf{z}_0 = E(\mathbf{x}_0)$ |
| 4:  $\mathbf{z}_0 = E(\mathbf{x}_0)$ | 4: $\mathbf{z}_t = \Omega_\Theta(\mathbf{z}_0, t)$ |
| 5:  $\mathbf{z}_t = \Omega_\Theta(\mathbf{z}_0, t)$ | 5: $\boldsymbol{\alpha} = \text{ReLU}(\mathbf{W}_\eta[\mathbf{h}_t, \zeta(\rho, \mu)] + \mathbf{b}_e)$ |
| 6:  $\mathbf{h}_t = \upsilon_{\theta_1}^{(t)}(\mathbf{z}_t, t), \mathbf{c})$ | 6: Do Intervention |
| 7:  $\boldsymbol{\alpha} = \text{ReLU}(\mathbf{W}_\eta[\mathbf{h}_t, \zeta(\rho, \mu)] + \mathbf{b}_e)$ | 7:  $\tilde{\boldsymbol{\alpha}} = [\alpha_1, \alpha_2, \ldots, \tilde{\alpha}_i, \ldots, \alpha_n]$ |
| 8:  $\hat{\mathbf{h}}_t = \text{Drop}_{\rho,\mu}(\mathbf{W}_\psi \boldsymbol{\alpha} + \mathbf{b}_d)$ | 8:   where $\tilde{\alpha}_i = \alpha_i \kappa$ |
| 9:  Do Gradient Descent Step on | 9: $\tilde{\mathbf{h}}_t = \text{Drop}_{\rho,\mu}(\mathbf{W}_\psi \tilde{\boldsymbol{\alpha}} + \mathbf{b}_d)$ |
| 10:   $\nabla_{\eta,\psi} \|\hat{\mathbf{h}}_t - \mathbf{h}_t\|_2^2 + \lambda \|\boldsymbol{\alpha}_i\|_1$ | 10: $\tilde{\mathbf{z}}_0 = (p_\Theta(\upsilon_{\theta_2}(\tilde{\mathbf{h}}_t, c)|\tilde{\mathbf{z}}_t)$ |
| 11: **until** converged | 11: $\tilde{\mathbf{x}} = D(\tilde{\mathbf{z}}_0)$ |

from $\mathbf{x}_0$ to $\mathbf{x}_T$ if we re-arrange equation 4 for $\mathbf{x}_t$ and assume that $\epsilon_\Theta^{(t)}(\mathbf{x}_t, \mathbf{c}) \approx \epsilon_\Theta^{(t-1)}(\mathbf{x}_{t-1}, \mathbf{c})$. This is known as DDIM inversion, and it fixes both our noise latents, as well as neural network representations through the forward and backward processes. This means we can investigate how their representations change under intervention without them changing by themselves, which aids representation learning. We denote DDIM inversion with $\Omega_\Theta(\mathbf{x}_0, t, \mathbf{c})$.

### 2.1.2 Latent Diffusion Models

Latent diffusion models extend diffusion models into the latent space of an image autoencoder, with encoder $E(\mathbf{x}) : \mathcal{X} \to \mathcal{Z}$ and decoder $D(\mathbf{z}) : \mathcal{Z} \to \mathcal{X}$. As such, the distribution that we approximate becomes $q(\mathbf{z})$ as opposed to $q(\mathbf{x})$. We use a latent diffusion model in the present work, and the notation henceforth reflects this.

## 2.2 H-Space

Diffusion model researchers have begun exploring the intermediate latent space, known as H-Space, at the bottleneck of the U-Net architecture used by $\epsilon_\Theta^{(t)}(\mathbf{z}_t, \mathbf{c})$ [6, 24–30]. Kwon et al. (2022) [24] showed that modifying activations within H-Space at specific timesteps, followed by continued denoising, enables isolated and interpretable changes to images. Importantly, the authors demonstrated that H-Space activations have three notable properties. First, they are robust to noise perturbations. Second, they are linear, allowing smooth interpolation between H-Space activations to create gradual changes in a concept. Third, they are consistent across images, enabling the use of the same H-Space direction to edit the same concept in different images. Given these properties and the reduced dimensionality compared to image-space, several works have developed H-Space modification frameworks for interpretable and isolated image editing [6, 24–30]. This makes H-Space a natural choice for the present work.

## 3 Method

We seek to learn a set of $L$ vectors $\{\boldsymbol{w}_i\}_{i=1}^L$, that reproduce H-Space activations when used in sparse linear combinations, such that each vector, $\boldsymbol{w}_i$ represents a distinct feature. We first decompose our noise prediction U-Net into its constituent parts:

$$\epsilon_\Theta^{(t)}(\mathbf{z}_t, \mathbf{c}) = \upsilon_{\theta_1}^{(t)}(\mathbf{z}_t, \mathbf{c}) \circ \upsilon_{\theta_2}^{(t)}(\mathbf{h}_t, \mathbf{c}) \quad \text{where} \quad \Theta = \{\theta_1, \theta_2\} \tag{5}$$

Where $\upsilon_{\theta_1}^{(t)}(\mathbf{z}_t, \mathbf{c})$ is the downward path of the U-Net, $\upsilon_{\theta_2}^{(t)}(\mathbf{h}_t, \mathbf{c})$ is its upward path and $\mathbf{h}_t$ is the H-Space representation of a given noisy latent $\mathbf{z}_t$, which is output by $\upsilon_{\theta_1}(\mathbf{z}_t)$.

### 3.1 The Channel-Aware SAE

In general, to learn our feature vectors $\{\boldsymbol{w}_i\}_{i=1}^L$ we may train an SAE with encoder weights $\mathbf{W}_\eta$ and decoder weights $\mathbf{W}_\psi$. By placing an $L_1$ penalty on our encoded, $\mathbf{h}_t$, we obtain $\boldsymbol{\alpha} = \mathbf{W}_\eta \mathbf{h}_t$,

Table 1: **Accuracy Scores on CelebA test set.** The columns denote the training dataset (CELEB-A and FFHQ Dataset).

| | Training Datasets | | |
| --- | --- | --- | --- |
| **Method** | **CelebA (Acc)** ↑ | **FFHQ (Acc)** ↑ | **Authors** |
| **H-Space SAEs (Ours)** | **0.80** | **0.80** | - |
| DiTi | 0.78 | 0.79 | Yu et al. (2022) [31] |
| PDAE | 0.79 | 0.79 | Zhang et al. (2022) [32] |
| Diff-AE | 0.77 | 0.79 | Preechakul et al. (2022) [33] |
| pSp-GAN | 0.78 | 0.78 | Richardson et al. (2020) [34] |
| $\beta$-TCVAE | 0.68 | 0.70 | Chen et al. (2018) [35] |

such that $\|\boldsymbol{\alpha}\|_0 \ll L$ and $\boldsymbol{\alpha} \in \mathbb{R}^L$. Our reconstruction of H-Space, $\hat{\mathbf{h}}_t$, is then a sparse linear combination of the columns of our decoder: $\hat{\mathbf{h}}_t = \sum_i^L \alpha_i \mathbf{w}_i$ where $\boldsymbol{\alpha} = [\alpha_1, \alpha_2, ..., \alpha_L]$, and $\mathbf{W}_\psi = [\mathbf{w}_1, \mathbf{w}_2, ..., \mathbf{w}_L]$. As such, we obtain our feature vectors $\{\boldsymbol{w}_i\}_{i=1}^L$. However, training SAEs directly on H-Space is computationally intensive. This is because U-Nets are convolutional neural networks, so $\mathbf{h}_t \in \mathbb{R}^{M \times K \times K}$ where $M$ is the channel dimension, $M > 1000$ and $K$ is a small number (e.g. 8). Let $D = MKK \approx 10^4$. Since SAEs expand the dimension of $D$, if we set the number of features $L = 2D$, we need $2 \times 10^8$ parameters for the encoder and decoder **each**. To alleviate this problem we introduce the *Channel-Aware SAE* by defining:

$$\bar{\mathbf{h}}_{t,i} = [\mathbf{h}_{t,i}, \rho, \mu] \quad i = 1, ..., M \quad \text{where} \quad \mathbf{h}_{t,i} \in \mathbb{R}^{K \times K} \tag{6}$$

Where $\rho$ and $\mu$ are scalar outputs of a neural network $\zeta_\gamma : \mathbb{R}^2 \to \mathbb{R}^2$ that models the channel dimension and the index of the image in the dataset corresponding to $\mathbf{h}_{t,i}$. We then treat all $\bar{\mathbf{h}}_{t,i} \in \mathbb{R}^{K \times K}$ as individual data points instead of directly modelling $M \times K \times K$ dimensional H-Space. This allows us to drastically reduce the computational cost of SAEs in H-Space, as instead of requiring around $2 \times 10^8$ parameters for both the encoder and decoder, we require approximately $2 \times 10^2$ since typically $K < 10$. The use of $\zeta_\gamma : \mathbb{R}^2 \to \mathbb{R}^2$ also allows this to be done flexibly.

### 3.2 Architecture and Training

As mentioned above we trained SAEs with a sparsity term, as such reconstructions of H-Space take the form:

$$\hat{\mathbf{h}}_t = \text{Drop}_{\rho,\mu}(\mathbf{W}_\psi \boldsymbol{\alpha} + \mathbf{b}_d), \quad \text{where} \quad \boldsymbol{\alpha} = \text{ReLU}(\mathbf{W}_\eta \bar{\mathbf{h}}_t + \mathbf{b}_e) \tag{7}$$

Where $\mathbf{b}_e$ and $\mathbf{b}_d$ are bias terms for the encoder and decoder, respectively, and $\text{Drop}_{\rho,\mu}$ denotes that we drop the last two dimensions corresponding to the channel dimension and image index. Our training objective takes the form:

$$\mathcal{L} = \frac{1}{N} \sum_{i=1}^N \|\hat{\mathbf{h}}_{t,i} - \mathbf{h}_{t,i}\|_2^2 + \lambda \|\boldsymbol{\alpha}_i\|_1 \tag{8}$$

Where $N$ is the size of the training set. Algorithm 1 shows the full training procedure.

### 3.3 Thresholding

Following training, we use thresholding to ensure that our features are used sparsely when reconstructing H-space. For a given $\mathbf{h}_t$, we compute its SAE representation $\mathbf{A} = [\boldsymbol{\alpha}_1, \boldsymbol{\alpha}_2, ..., \boldsymbol{\alpha}_M]^T$ where $M$ is the channel dimension of $\mathbf{h}_t$. We then compute:

$$\bar{\mathbf{A}} = [\text{TopK}(\boldsymbol{\alpha}_1), \text{TopK}(\boldsymbol{\alpha}_2), ..., \text{TopK}(\boldsymbol{\alpha}_M)]^T \tag{9}$$

Where $\text{TopK}(\cdot)$ returns the highest $k$ values. We then define $\tau = \min(\bar{\mathbf{A}})$ and set all values of the original matrix $\mathbf{A}$ to 0 if they are below $\tau$. Taking the minimum of $\bar{\mathbf{A}}$ represents a trade-off between sparsity and data reconstruction ability.

Table 2: **Evaluation Metrics for Different Methods on CELEBA, CELEBA-HQ, and FFHQ Datasets.** Metrics include explained variance (EV), cosine similarity, mean squared error (MSE), and mean absolute error (MAE).

| | CELEBA | | | | CELEBA-HQ | | | | FFHQ | | | |
|---|---|---|---|---|---|---|---|---|---|---|---|---|
| | EV ↑ | Cosine ↑ | MSE ↓ | MAE ↓ | EV ↑ | Cosine ↑ | MSE ↓ | MAE ↓ | EV ↑ | Cosine ↑ | MSE ↓ | MAE ↓ |
| **H-Space SAE (Ours)** | **0.90** | **0.95** | **1.81** | **1.07** | **0.90** | **0.95** | **1.73** | **1.04** | **0.89** | **0.95** | **1.85** | **1.08** |
| PCA | 0.54 | 0.83 | 6.04 | 1.91 | 0.57 | 0.84 | 5.75 | 1.86 | 0.54 | 0.83 | 6.08 | 1.91 |
| $\beta$-VAE (16 dim) | -3.5 | 0.42 | 15.8 | 3.1 | -3.4 | 0.43 | 14.3 | 2.9 | -3.4 | 0.39 | 16.2 | 3.1 |
| $\beta$-VAE (512 dim) | -3.2 | 0.43 | 15.6 | 3.0 | -3.1 | 0.47 | 13.7 | 2.8 | -3.1 | 0.43 | 15.6 | 3.0 |
| Random | - | 0.01 | $7.60\times10^5$ | $6.70\times10^2$ | - | 0.01 | $6.39\times10^5$ | $6.12\times10^2$ | - | 0.01 | $7.60\times10^5$ | $6.70\times10^2$ |

## 3.4 Interventions on H-Space

### 3.4.1 From Image Space to SAE Space

To evaluate the concepts that our features, $\{w_i\}_{i=1}^L$, correspond to, we perform *interventions* on H-Space and observe how our intervention modifies the original image. To achieve this, we need to transition from image-space to SAE Space, by following a four-step procedure. 1) We encode an image $\mathbf{x}$ with $E(\mathbf{x})$ to obtain our latent $\mathbf{z}_0$. 2) We then run $\mathbf{z}_0$ through DDIM Inversion, $\Omega_\Theta(\mathbf{z}_0, t, \mathbf{c})$, to obtain $\mathbf{z}_t$ our noisy latent. 3) We then transition into H-Space by placing $\mathbf{z}_t$ through the downward path of our U-Net $\upsilon_{\theta_1}(\mathbf{z}_t, \mathbf{c})$ and append the channel dimension along with the image index to obtain $\bar{\mathbf{h}}_t$. 4) Finally, we transition into SAE space by running $\bar{\mathbf{h}}_t$ through our SAE encoder to obtain $\boldsymbol{\alpha}$.

We may then intervene on H-Space. By modifying a single $\alpha_k \in \boldsymbol{\alpha}$, we up weight the feature $\mathbf{w}_k$ because $\hat{\mathbf{h}}_t = \text{Drop}_{\rho,\mu}(\sum_k^L \alpha_k \mathbf{w}_k + \mathbf{b}_d)$. Generally, we scale an embedded component $\alpha_k$ to produce our modified SAE representation $\tilde{\boldsymbol{\alpha}}$, such that:

$$\tilde{\boldsymbol{\alpha}} = [\alpha_1, \alpha_2, \ldots, \tilde{\alpha_k}, \ldots, \alpha_L] \quad \text{where} \quad \tilde{\alpha}_k = \alpha_k \kappa \quad \kappa \in \mathbb{R} \tag{10}$$

### 3.4.2 From SAE Space back to Image Space

To observe the effect of our intervention we must return from SAE-space back to image-space by following a three-step procedure. 1) We place $\tilde{\alpha}$ through the SAE decoder and reconstruct H-Space by computing $\tilde{\mathbf{h}}_t =$. 2) We then run $\tilde{\mathbf{h}}_t$ through the upward path of the U-Net $\upsilon_{\theta_2}(\tilde{\mathbf{h}}_t, t, \mathbf{c})$ to obtain $\tilde{\mathbf{z}}_t$. 3) Finally, $\tilde{\mathbf{z}}_t$ is run through the forward process, $p_\Theta(\tilde{\mathbf{z}}_0|\tilde{\mathbf{z}}_{t:1})$ to obtain our modified $\tilde{\mathbf{z}}_0$, which is decoded with $D(\tilde{\mathbf{z}}_0)$ to obtain our modified image $\tilde{\mathbf{x}}$.

To infer which concept $\mathbf{w}_k$ corresponds to, we then observe the change between our original image $\mathbf{x}$ and the modified image $\tilde{\mathbf{x}}$. To infer which neurons within H-Space correspond to these features, we take the difference between the original $h_t$ and the modified $\tilde{h}_t$ Note that this occurs at a single time step in the denoising process and modifies a single SAE latent vector component.

For our experiments, we chose $t = 0.7$ where $T = 1000 =$ total denoising steps because previous work has shown that it is early enough in denoising to change concepts effectively but late enough that the concept changes are robust [6, 28]. Algorithm 2 shows the algorithm in full. Note that fully flattened $\mathbf{A}$ still has approximately $1000 \times 8 \times 8$ dimensions.

## 4 Experiments

### 4.1 Datasets

We applied our H-Space SAEs to three publicly available datasets: Celebrity Faces Attributes (CELEB-A), consisting of 200,000 images each labelled with 40 binary attributes [36]; CELEBA-HQ, the higher resolution variant of CELEB-A, comprising 30,000 images [37] and Flickr-Faces-HQ (FFHQ), which includes 70,000 images of faces from the image website Flickr [38].

### 4.2 Training Set-up

We used Stable Diffusion 2.1, [39] as our pre-trained latent diffusion model. The dimension of H-Space in this model is $1280 \times 8 \times 8$, so $\bar{\mathbf{h}}_{t,i}$ was 66 dimensional. We chose a hidden size of 512 for $\alpha$ when training on CELEB-A and FFHQ, and a hidden size of 256 for CELEBA-HQ and AFHQ. Our hardware consisted of 2 Nvidia A100s, and an Nvidia RTX 4090.

| Original | Modified | Inverted | Original | Modified | Inverted |

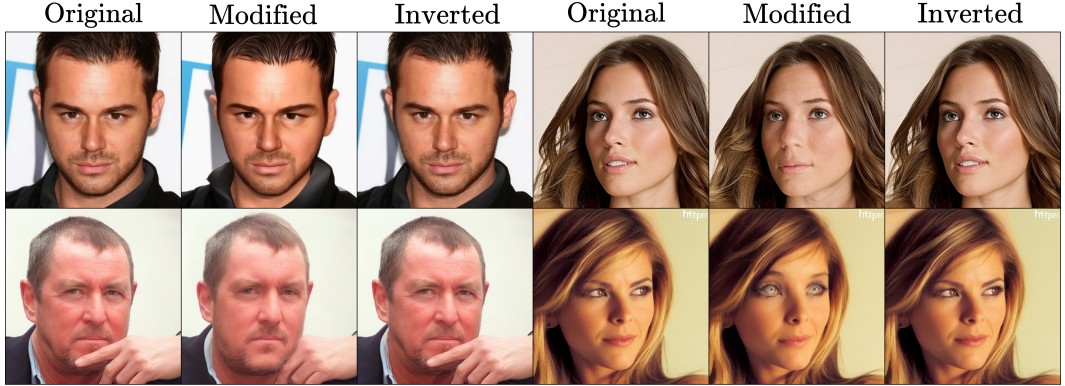

Figure 3: **Counterfactual Consistency Experiment results.**

## 4.3 Quantitative Assessment

### 4.3.1 4.3.1 Linear probes

To assess the quality of the representations produced by our method, we followed a similar protocol to Yue et al. (2024) [31]. We trained one SAE on H-Space representations from the CELEBA train split and another SAE on the FFHQ dataset. Once these SAEs were obtained, we trained linear probes to classify the 40 binary facial attributes based on SAE representations of the CELEBA training set. We then evaluated our linear models on the CELEBA test set. Our baselines fell into two categories: i) traditional unsupervised representation learning techniques ($\beta$-TCVAE [35] and pSp-GAN [34]), and ii) diffusion-based methods (DiffAE [33], PDAE [32] and DiTi [31]). To ensure the input to the probes were of the same dimension, we used the 512-dimensional variants of the baseline methods. For our SAEs we took the top 512 components from, our flattened out representations, $\mathbf{A} = [\boldsymbol{\alpha}_1, \boldsymbol{\alpha}_2, ..., \boldsymbol{\alpha}_M]$.

### 4.3.2 Reconstruction

We also evaluated our SAEs' ability to perform out-of-distribution generalization by assessing their ability to reconstruct H-Space representations on unseen data. We trained SAEs on the CELEBA train set, CELEBA-HQ, and FFHQ. We then evaluated the performance of these models on the CELEBA test set. The metrics used for evaluation were explained variance, mean squared error, mean absolute error, and cosine similarity. We compared our model's results to the following baselines: two non-linear baselines ($\beta$-VAE-512 dimensional and $\beta$-VAE 16 dimensional), a linear baseline: principal component analysis (PCA), a random baseline: 2 normally distributed matrices were for the encoder and decoder respectively, which were then used to reconstruct H-Space representations.

## 4.4 Qualitative Assessment

To qualitatively assess our method, we take the top 10 components from our flattened out SAE representations and intervene on their values. Additionally, we demonstrate the results of interpolating between different images in SAE space. We also conduct a counterfactual consistency experiment, where we modify a particular feature within an image, and then change it back to see if other features change, the baseline we compare to is text based image editing. Finally, to mechanistically interpret our SAE representations, we analyze which dimensions in SAE space typically have the highest activation values on CELEBA-HQ. We then examine which class examples maximally activate these top $k$ activations.

## 4.5 Results

### 4.5.1 Representation Learning Capacity

Table 1 compares the CELEBA test set accuracy achieved by linear probes trained on our SAEs representations to both conventional and diffusion-based unsupervised representation learning meth-

ods. We achieve competitive accuracy scores with the state-of-the-art diffusion based representation learning methods, with a marginal performance gain; this demonstrates the high quality of the representations that our SAEs produce. As expected, $\beta$-VAE performs poorly with high-resolution facial data, as it was developed to work on lower resolution synthetic datasets.

Table 2 displays how our method compares to both linear (PCA) and non-linear ($\beta$-VAE) dimensionality reduction techniques, on the task of reconstructing unseen data. We significantly outperform all baselines across all metrics, demonstrating the capacity of our SAEs to effectively capture the variance within H-Space representations.

After using our thresholding technique, we achieve an average of 95 % sparsity of activations (only 5% of SAE representations are active) within SAE space across SAEs trained on all datasets. Despite this, we still outperform our baselines, which suggests that we recover features that recover our H-Space activations when sparsely combined. As expected, the 512 latent dimensional network generally outperforms the 16-dimensional variant.

### 4.6 Concept Intervention

Figure 2 displays images where we intervene on one of the top 10 components within SAE space. After scaling a single one of these components we observe interpretable image edits. We also denote the number of neurons which changed within H-Space before and after intervention. For example, in the image in the top left of the figure 2 after scaling component 57512 of $\mathbf{A}$ by 750 we can observe that the mouth closes.

Approximately, 3683 ($0.009\%$) of neurons changed within H-Space before and after the intervention, and this demonstrates how localization of the changes made by our method.

#### 4.6.1 Counterfactual Consistency

For an intervention on a concept to be localized, it must be invertible, meaning the operation causing the change can be reversed to nullify the effect. We demonstrate that our SAEs recover this property in Figure 3. For example, in the top right corner of the figure, we scale component 1678 by 1000 and see a modification to the individual's age. To reverse the modification, we compute the SAE representation of the modified image and divide component 1678 by 1000. This demonstrates the invertibility of the features learnt by our SAEs which speaks to the robustness of the representations learnt.

## 5 Discussion, Limitations and Future Work

In this work, we presented the H-Space SAE. This technique shows promise as a tool for enhancing the safety of image-generative models by improving control and interpretability in diffusion models. By intervening on a single SAE component at a single diffusion timestep, we obtain interpretable changes to image generations. This enables targeted interventions that may be used to prevent unsafe or biased content generation, such as offensive images or deep fakes.

We also introduced the Channel-Aware SAE in order to drastically increase the computational efficiency of SAEs applied to diffusion models. Such fine-grained and computationally efficient control could be crucial in real-time AI systems to ensure ethical outputs and reduce risks of misuse, contributing to safer deployment of generative models. Notably, this work establishes a connection between safety techniques used in language modelling and image-generative modelling. An increase in cross-modality safety techniques may lead to a deeper understanding of how neural networks work more fundamentally, which, in turn, may lead to greater control over the model outputs of both modalities.

A limitation of the current work is that the specific neurons responsible for particular outputs in the diffusion models have not been systematically identified, unlike the mechanistic interpretability applied to language models to locate circuits. Future work should employ the present method to identify how changes in SAE space map onto specific neurons within the diffusion model.

Additionally, we only examined feature manipulation at a single diffusion time step. Future work should explore the effects of interventions across multiple timesteps, which could provide a more comprehensive understanding of how features evolve throughout the diffusion process and enable

finer control over outputs. Expanding on these areas could significantly improve the precision and scope of this technique in both interpretability and practical application.

# 6 Related Work

Recent efforts to explore and manipulate H-Space representations in diffusion models have led to various advancements. Kwon et al. (2022) were the first to modify H-Space activations, inspiring techniques like Epstein et al. (2023), who used self-guidance for meaningful representation extraction, and Jeong et al. (2023), who injected content into H-Space using text prompts. Similar approaches by Haas et al. (2023) have also been proposed, all relying on supervisory signals to control model behavior [25, 28, 30]. In contrast, the present method automatically identifies interpretable directions without external guidance.

Additionally, Yong et al. (2023) applied the pullback metric from Riemannian geometry to decompose H-Space, enabling the discovery of interpretable features [6]. However, their approach induced large-scale changes, whereas the present method focuses on localized, precise interventions. Other related work includes the language modelling SAE literature [13, 22, 40]. For example Cunningham et al. (2023) [40] demonstrate that SAEs can disentangle complex data, uncovering interpretable neurons in language models, which parallels efforts in diffusion models to identify and control specific features.

# 7 Potential Negative Societal Impact

While our method enhances control over diffusion model outputs, it may also facilitate misuse in sensitive applications. For instance, targeted modifications could be exploited for unethical purposes, such as deepfake creation, image alteration without consent, or propagating harmful stereotypes. Additionally, increased accessibility to such manipulation tools may raise privacy concerns and exacerbate bias if applied without adequate safeguards. Proactively addressing these risks through ethical guidelines and safeguards is essential to prevent potential harm and misuse.

# 8 Acknowledgements

This work was supported by funding from the Engineering, and Physical Sciences Research Council (EPSRC), the UCL Centre for Doctoral Training in Intelligence, Integrated Imaging in Healthcare (i4health) and the Motor Neuron Disease (MND) Association.

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
