# OpenReview forum: "H-Space Sparse Autoencoders"
_NeurIPS.cc/2024/Workshop/SafeGenAi — SafeGenAi Poster_

### Official Review · Reviewer_ReSa · 2024-10-09
**Reasonable paper**

**Rating:** 6
**Confidence:** 1

**Review:**

Due to paucity of time, a full review could not be conducted.

Are the results in Table 1 and 2 statistically significant?

---

### Official Review · Reviewer_WYCZ · 2024-10-09
**Overall, an interesting work which has a large space to improve to provide a richer understanding**

**Rating:** 6
**Confidence:** 3

**Review:**

Paper introduces an interesting concept of H-space SAEs for intervening in diffusion model generation to show single neuron intervention provides significant feature changes in the generated output. The work provides valuable future directions of improvement, such as the choice of $t=0.7T$. It would be interesting to see the change in results as the choice of t moves in both directions, and also a richer understanding of the interventions.

Strengths:
 - Interesting idea and links to work in Language modelling
 - Good quantitative and qualitative results
 - posed and solved the problem of computational limitations in Hspace of Unets

Weaknesses:
- 3.4.1 and 3.4.2 are a little bit hard to follow and the algorithms 1 & 2 could benefit from more descriptive information about what happens at each line as otherwise the reader needs to keep in mind a lot of notation to follow.
- There doesn't seem to be an explanation on why the middle of the Unet is used as the Hspace, this may be common knowledge in the specific area but could be beneficial to state why.
- The intervention in the Hspace seems a little bit ad hoc and could do with explaining how and why specific interventions are chosen, if there are any.


Minor Issues:
- line 79: deep diffusion implicit Models should be Denoising Diffusion Implicit Models
- Sec 3.4.2:  A few mistakes in this section, unfinished eq (L.152), forward process (L.153)

---

### Official Review · Reviewer_qtfg · 2024-10-10
**Paper Review**

**Rating:** 7
**Confidence:** 3

**Review:**

The paper presents H-Space Sparse Autoencoders (SAEs) as a method to improve the interpretability and safety of diffusion models. It focuses on leveraging the latent space within the U-Net architecture of diffusion models, known as “H-Space,” to enable controllable and interpretable modifications to generated images. Notable contributions include the introduction of the Channel-Aware SAE and interpretable modification techniques.

Both quantitative and qualitative assessments are conducted on multiple datasets (CELEB-A, CELEBA-HQ, and FFHQ), where the method demonstrates competitive performance, slightly surpassing diffusion-based models such as Diff-AE and DiTi, as well as traditional models like β-TCVAE. Additionally, the paper’s use of thresholding and activation sparsity illustrates how SAEs can isolate specific features with high precision.